# Large-Scale Cellular Vehicle-to-Everything Deployments Based on 5G—Critical Challenges, Solutions, and Vision towards 6G: A Survey

**DOI:** 10.3390/s23167031

**Published:** 2023-08-08

**Authors:** Dániel Ficzere, Pál Varga, András Wippelhauser, Hamdan Hejazi, Olivér Csernyava, Adorján Kovács, Csaba Hegedűs

**Affiliations:** 1Department of Telecommunications and Media Informatics, Budapest University of Technology and Economics, 2 Magyar Tudósok krt., H-1117 Budapest, Hungary; ficzere@tmit.bme.hu (D.F.); hegedus.csaba@tmit.bme.hu (C.H.); 2Department of Networked Systems and Services, Budapest University of Technology and Economics, 2 Magyar Tudósok krt., H-1117 Budapest, Hungary; wippelhauser@hit.bme.hu (A.W.); hhejazi@hit.bme.hu (H.H.); 3Department of Broadband Infocommunications and Electromagnetic Theory, Budapest University of Technology and Economics, 2 Magyar Tudósok krt., H-1117 Budapest, Hungary; csernyava.oliver@edu.bme.hu; 4Department of Automation and Applied Informatics, Budapest University of Technology and Economics, 2 Magyar Tudósok krt., H-1117 Budapest, Hungary; adorjan.kovacs@aut.bme.hu

**Keywords:** cellular V2X, use case, 5G, 6G, cellular positioning, deployments, 3GPP

## Abstract

The proliferation of fifth-generation (5G) networks has opened up new opportunities for the deployment of cellular vehicle-to-everything (C-V2X) systems. However, the large-scale implementation of 5G-based C-V2X poses critical challenges requiring thorough investigation and resolution for successful deployment. This paper aims to identify and analyze the key challenges associated with the large-scale deployment of 5G-based C-V2X systems. In addition, we address obstacles and possible contradictions in the C-V2X standards caused by the special requirements. Moreover, we have introduced some quite influential C-V2X projects, which have influenced the widespread adoption of C-V2X technology in recent years. As the primary goal, this survey aims to provide valuable insights and summarize the current state of the field for researchers, industry professionals, and policymakers involved in the advancement of C-V2X. Furthermore, this paper presents relevant standardization aspects and visions for advanced 5G and 6G approaches to address some of the upcoming issues in mid-term timelines.

## 1. Introduction

Fifth-generation communication networks are designed to support novel advanced use cases, facilitating technologies like massive machine-to-machine communication, ultra-reliable low-latency communication (URLLC) services, or multi-access mobile edge computing solutions. One of its major use cases is the V2X—vehicle to anything—communication. In the automotive industry, V2X communication is a novel technology addressing many critical safety use cases; thus, the security and safety of the potential V2X networks is essential. In the automotive world, the faulty operation of systems can easily cause physical harm. Thus the quality of the communication channel is crucial. Apart from that, V2X communication also features different requirements than usual due to the peculiar design principles of V2X communication.

The 5GAA has three studies about several use cases, where these reports present use case descriptions and service level requirements [1,2,3]. In addition, the 5GAA has several other very high-quality studies and reports in this field, such as [4], which introduces system enablers and best practices for V2X services. Similarly, there are documents with 3GPP support about V2X service requirements. These requirements are identified by taking into account the V2X service requirements defined by ETSI ITS and US SAE. The specification includes requirements of safety and non-safety aspects [5].

Ref. [6] provides an overview of V2X use cases, their potential outcomes, and a comparison between the potential requirements based on their categorization, i.e., latency, message size, mobility performance, response time, message frequency, anonymity, integrity, reliability, and communication. In contrast, Ref. [7] provides a survey of the latest V2X use cases, including requirements and various 5G-enabling technologies under consideration for vehicular communications. Subsequently, we first provide an interesting mapping between the three 5G pillars and V2X use case groups. While Ref. [8] outlines a series of key enabling technologies from a range of domains, such as new materials, algorithms, and system architectures, Ref. [9] provides a comprehensive overview on 5G-MEC testbeds for V2X applications. Aiming for truly intelligent transportation systems, we envision that machine learning (ML) will play an instrumental role in advanced vehicular communication and networking. Furthermore, Ref. [10] presents novel C-ITS use cases that utilize machine learning. The primary objective of this paper is to demonstrate the potential of artificial intelligence in enhancing C-ITS applications. This paper shows that C-ITS applications are built on vehicle-level, intervehicle-level, and infrastructure-level applications as well.

There are several papers and studies and research about the requirements and enablers of V2X use cases—including [8] as a survey on use case requirements, ref. [11] as a survey on V2X slicing, ref. [12] as a survey on V2X cybersecurity mechanisms, or ref. [13] on C-V2X for Level 5 autonomous driving from 3GPP standards’ viewpoint, among others. Still, there is not much work that analyzes the potential controversies, obstacles, and challenges in this field. The objectives of this research paper are as follows:Identify, analyze, and extend the understanding of the key challenges related to the large-scale deployment of 5G-based cellular vehicle-to-everything (C-V2X) systems.Pinpoint obstacles and controversies in the cellular V2X standards caused by special requirements.Introduce the main definitions and standards in connection with C-V2X and explore RAN security and positioning.Highlight influential C-V2X projects that have a significant impact on the widespread adoption of this transformative technology.

By thoroughly examining these challenges, our research contributes to a comprehensive understanding of the complexities involved in the deployment of 5G-based C-V2X systems. This paper provides valuable insights for researchers, industry professionals, and policymakers engaged in advancing connected transportation systems. By identifying and analyzing the critical challenges and extending the discussion to encompass standards and influential projects, this paper aids in the successful deployment and wide-scale adoption of 5G-based C-V2X systems.

The paper is structured as follows: Section 2 provides a brief introduction and the main aspects of the 5G-based C-V2X use case requirements. Section 3 will introduce the evolution of cellular V2X technologies, the secrecy aspects of MIMO antennas, and 5G-based localization. Section 4 summarizes several research projects and consortia works in recent years. The main controversies, obstacles, and challenges found are presented in Section 5. Section 6 briefly summarizes the main concepts foreseen to affect 6G-V2X, while Section 7 concludes the paper. This structure is reflected in Figure 1. Furthermore, Table 1 summarizes the references used for this paper.

## 2. Requirements of 5G-Based V2X Use Cases

This section explores the requirements of vehicle-to-everything (V2X) use cases focusing on cellular communication. We do not discuss the C-V2X requirements in great detail as there are many other research papers, studies, and recommendations covering this topic. However, a brief overview of the C-V2X requirements is essential for the discussion and understanding of the enablers, recent research, obstacles, and challenges of the C-V2X, which are the core and novelty of this paper.

The V2X use cases are mainly safety-relevant use cases. Thus, high reliability and availability are the most crucial requirements. In order to design such a system, the V2X protocols are designed to support distributed operation [7,15,27]. Among other implications, this design principle ensures that a defective node has only a minor effect on the operation of the whole system. The distributed operation implies that the processing of the shared data and the decisions made (based on the data) are taken in the individual vehicles. This also means that the communication basically shares awareness data. These data include the status, perception, or intention of the ego entity or the status of the infrastructure. The data are broadcasted periodically within a local geographic scope. A certain level of packet loss is usually acceptable since the shared information is repeated (and updated) periodically. This operation also implies that the communication is signaling-typed rather than session-typed [28]. The V2X applications rely on standards specified by ETSI or SAE, depending on the region. In addition to this, the 5GAA [1,2,3] defines requirements for the applications.

### 2.1. QoS Requirements

#### 2.1.1. Throughput Requirements

One of the most fundamental requirements of any use case is the throughput or data rate. The 3GPP defines data rate requirements for the different V2X scenarios on a per-user level. These recommended data rate values are achievable, but there are quite a few fully operational use cases or measurement campaigns where requirements such as the data rate are comprehensively examined. The main challenge would be the use case behavior in crowded areas and how the overall capacity and data rate requirements can be reduced in such transportation scenarios [19,28,29].

#### 2.1.2. Latency Requirements

Usually, the latency requirements are defined somehow as the time between the sending and the receiving of a specific packet. In the case of V2X applications—or any application—latency requirements are defined as the maximum tolerable elapsed time from the instant a data packet is generated at the source application to the instant it is received by the destination application. So, the latency requirements towards the cellular network can be different from the end-to-end application latency requirements. The latency has many aspects in regular networks, including the jitter, the average latency, or the chances of getting very high latency values [5,28]. Furthermore, in V2X communication, we instead opt for information latency. This means that the information in a packet is frequently updated, and the receiver is always interested in the most recent information available. Consequently, we might drop a packet and rely on the reception of the next packet instead of spending valuable time on the retransmission of an old packet. Detailed latency requirements are defined in [1,2].

#### 2.1.3. Packet Drop

In regular stream-like applications, the packet drop is a crucial performance indicator [28]. The exchange of medium- or large-sized encoded data or data protected by stream encryption methods is usually useless in the case that some of the packets are missing from the data stream. The packet drop might also increase the performance requirements against clients, like their TCP buffer size. Contrary to this, V2X facilitates compact data segments, which can travel in single packets. Since V2X applications are primarily interested in the most recent data, the packet drop parameter partially loses its importance and eventually becomes an access technology-specific trade-off between the packet drop, latency, and bandwidth. The adjusted parameter can be the message frequency. The evaluation criteria of [120] is eye-opening since it highlights well that the goal of V2X communication is not to limit the number of packet drops but to minimize the age of the information.

#### 2.1.4. Jitter Requirements

Jitter is another fundamental component of the QoS. It is a form of packet delay variation and is closely related to latency characteristics. Low jitter can be easily achieved while increasing latency with some form of buffer or shaper. However, the real challenge is to achieve low jitter with a low latency, as many real-time applications require it, including many V2X applications. The 3GPP standards [28,29] do not address jitter explicitly in their V2X use case requirements, but the formulation of the latency requirements implicitly includes constraints on jitter as “Max end-to-end latency (ms)”. Also, there are several C-V2X use cases that require low max end-to-end latency—sub 10 ms—such as sensor information sharing for a high degree of vehicle automation, remote driving, emergency trajectory alignment, and platooning. However, few comprehensive measurement campaigns or research studies focus on the jitter characteristics of C-V2X use cases [18].

### 2.2. The Security Model

In regular applications, we mostly handle the security questions by encrypting the message exchanges. This ensures that the content of the messages cannot be eavesdropped on. The currently available V2X message standards, like those of the European Telecommunications Standards Institute (ETSI), have established standardization for the Cooperative Awareness Basic Service and Decentralized Environmental Notification Basic Service. These standards aim to facilitate vehicular safety and enhance traffic efficiency by providing continuous status updates on nearby vehicles and asynchronous event notifications [17,25,26]. The specifications for these standards encompass the packet formats for both the Cooperative Awareness Message (CAM) and the Decentralized Environmental Notification Message (DENM) [26]. The CAM and DENM typically operate only with a signature; no encryption is involved in the message profiles. It is because those messages are only broadcasted in the close physical neighborhood of the sender entity, and all the subsequent entities need to process that data. However, the authenticity of the message has to be protected, which is handled by signatures. A public key infrastructure architecture was also defined in order to enhance interoperability. It is also worth mentioning that the traceability of the vehicles also has to be considered. This is handled by so-called pseudonymity protection techniques, which ensure that all unique identifiers of the vehicles are exchanged regularly. Nevertheless, these statements are only valid for standardized message services, which typically enhance the awareness of the traffic participants. Other use cases, especially centralized use cases, like digital twins or remote driving, are not discussed here. In those cases, the security model is different. This paper does not introduce and discuss security standards in a comprehensive manner. It is beyond the scope of this paper, as several other works focus on the security aspects of cyber–physical systems [14,22].

### 2.3. Availability

A high availability is essential in V2X because it directly affects the safety and efficiency of the transportation system. V2X services rely on real-time communication between vehicles, infrastructure, pedestrians, and other devices in the environment. The downtime and traffic interruptions of the communication network can cause delays, accidents, and other safety concerns. Therefore, high availability is critical to ensure that V2X services function reliably and effectively. The security and safety requirements of the V2X traffic imply that the communication system shall also work without a centralized infrastructure. This ensures availability in areas where the coverage is insufficient and it also mitigates the risk of a central system failure [7,15,30].

### 2.4. Positioning

The V2X applications usually cover low-latency, safety-critical use cases where the distance between the vehicles is relatively low. The use cases mostly investigate situations where the ego vehicle is/will be in some kind of relationship with the remote event or object [16,21]. Such an event can be a traffic light change, an accident, or road works. The objects can be vehicles, emergency vehicles, or pedestrians, for example. Since the relation is the most crucial in these cases, the relative position is more important than the absolute position. This differs from most of the classical use cases, like conventional navigation applications or geo-fencing-based warning applications. In those applications, the absolute position of the client is the most important. When it comes to positioning with 5G and beyond technologies, there are various methods [21] available for V2X purposes as well.

### 2.5. Power Consumption

In regular applications, the power consumption of a radio chip is crucial. This is mostly due to the limited size and mobility requirements of the user equipment. As opposed to this, the main participants of the V2X communication—Road Side Units and On-Board Units—have a constant power supply, which means that the power consumption of the V2X chips is not a relevant factor by design. Thus, the V2X radios are listening in most of the time and transmitting if that is required, but they are not switched to power-saving mode, where reception is not possible [20,23,24].

### 2.6. Typical Traffic Format

The V2X ecosystem assumes a distributed system due to various requirements. Thus, the messaging formats used are standardized. Those messages are encoded efficiently into relatively small packets. The size of the packets depends on the encoded information. Not all of the information is encoded in all packets; some segments are periodically included. The security certificates are also only encoded in some of the packets (typically in every 10th). This results in frequently changing packet sizes. Various studies define the packet size or the data volume for a certain period of time and the sending frequency for different scenarios depending on the traffic situation, vehicle density, and transport environment [1,2,28].

## 3. 5G Technology Enablers for V2X Use Cases

This section gives a brief introduction to the evolution of V2X communications. Figure 2 gives an overview of the timeline and connection between the 3GPP releases, their main features, and the driving V2X use cases. It is also worth noting that this is not a full in-depth overview, but the key milestones are mentioned. Moreover, it focuses on the RAN features and capabilities of 5G, including enabler features, secrecy aspects of MIMO antennas, and 5G-based positioning.

### 3.1. The Evolution of V2X Communications

The efforts already taken in 5G and the remaining research topics are summarized in great detail in [8,65]. Ref. [65] outlines the key features of the standard, including the use of a new frequency band and advanced signal processing techniques to support a low latency, high reliability, and high throughput communication. The authors also present models for evaluating the performance of the standard under various scenarios, such as urban and highway environments. Ref. [8] identifies the enabling technologies, challenges, and opportunities associated with the implementation of 6G networks in V2X communication. The paper discusses the technological advancements required to achieve these benefits, such as millimeter-wave communication, massive MIMO, and advanced beamforming techniques.

#### 3.1.1. WiFi-Based Technologies

The first dedicated V2X communication solution (released in 2010) was the 802.11p protocol [52], which was developed from the popular 802.11 standard family. The 802.11p is mostly a modified version of the 802.11a standard, tailored to vehicular communication. This means that the channel access model (CSMA/CD) stayed the same as the WiFi technologies. However, certain standard parameters were changed in order to support the high speeds and the fast-changing topology of the nodes, which is specific to V2X communication. Using the WiFi technology also implies that no concurrent transmission is available, OFDM modulation is used, and there is loose asynchronous time synchronization. In the WiFi-based V2X networks, the 802.11bd standard is actively developed. This standard will be backward compatible with 802.11p and outperforms it in many areas [122].

#### 3.1.2. 3GPP Release 14

In 2017, 3GPP Release 14 was introduced. This set of standards includes V2X communication, which is basically an extension of the Release 12 LTE Proximity Services used for ad hoc direct communication services. The communication methodology is similar to the LTE uplink communication of the Release 14 standard. This implies that the same subchanneling mechanisms are applied, which requires tight time synchronization, SC-FDM modulation, and turbo coding. The Release 14 communication defines Mode 3 and Mode 4 transmission scheduling modes. In Mode 3, the cellular network is responsible for resource management (subchannel assignment). Mode 4 defines a distributed operation, which is essential in the V2X world due to safety reasons, as depicted in [59]. In Mode 4, the tight time synchronization requirement was solved via precise GNSS signals. This also means that the Release 14 distributed communication is not operable without a GNSS signal [8,30,58].

#### 3.1.3. 3GPP Release 15

Release 15 (released in 2019) is basically an enhanced version of Release 14; the communication did not change in principle [30,74]. The vehicular communication is now referred to as eV2X. New requirements were specified in [5,28], such as vehicle platooning, advanced driving, extended sensors, and remote driving, in order to enhance support for V2X scenarios. While keeping the full backward compatibility with Release 14, Release 15 introduced the following advancements [123]:Carrier aggregation support for Mode 4 communication (Mode 3 CA was already supported in Release 14).Support for 64 QAM.A reduction in the maximum time between packet arrival at Layer 1 and the resource being selected for transmission.Radio resource pool sharing between Mode 3 and Mode 4 UE.Transmission diversity.Other RF and RRM requirements.

#### 3.1.4. 3GPP Release 16

Release 16 was introduced in 2021. It starts to specify the V2X relevant core network elements—namely the V2X router functionality—and the new radio (NR)-based sidelink communication [64]. Within the core network, requirements were developed in relation to the vehicular communication quality of service support. The main features of Release 16 and later releases are summarized in Figure 3. The work concluded that a standardized interface is needed in order to negotiate the QoS requirements and changes in the QoS among the client and the V2X application servers. Architecture enhancements for V2X services were also introduced. This included interworking between EPS V2X and 5GS V2X, the PC5 and Uu reference points, their various configuration parameters, and alternative service requirements. The V2X application-specific layer and the V2X application support layer were defined in the application layer. The latter contains the Service Enabler Architecture Layer (Service Enabler Architecture Layer—SEAL) and the V2X Application Enabler Layer (V2X application enabler—VAE). The 5G V2X with an NR sidelink is a close relative of the LTE-based V2X sidelink communication; however, it is not backward compatible with the legacy system [57]. The 5G introduces two resource allocation modes, Mode 1 and Mode 2 (similarly to the LTE V2X’s Mode 3 and Mode 4). In Mode 1, the resource allocation is performed by gNB and in Mode 2 the UE selects the resources autonomously [66]. The time synchronization issue is solved via GNSS signals, just like in the predecessor standards [60,68].

Apart from the dedicated interface designed for V2X communication, there were other efforts as well in the scope of the 5G standards, which could be important from the V2X perspective. Such efforts are MEC, URLLC, and uM2M communication:Multi-access mobile edge computing is expected to enable low-latency and highly reliable service provisioning since it is able to deploy services near the vehicles themselves.Ultra-reliable low-latency communication (or ultra-reliable machine-type communication) drives mission-critical information exchange. One of its use cases is remote driving, but, apart from that, the URLLC combined with the MEC could serve as a foundation for many centralized V2X applications.

#### 3.1.5. 3GPP Release 17

Release 17 investigates the enhanced support of V2X operation for pedestrian UE (i.e., UE for vulnerable road users), e.g., V2X communication with power efficiency, according to the vehicular services requirements defined in TS.22.185. [5] and TS.22.186. [28].

Pedestrian UE often has a lower battery capacity and limited radio capability compared to vehicular UE, which can affect its power consumption and ability to send/receive V2X messages. To address this, the use of Discontinuous Reception (DRX) has been identified as a mechanism for power saving in sidelink communications. The objective is to support power-saving requirements for pedestrian UE in V2X and other use cases. For example, periodic broadcasts of road safety messages may require different DRX cycles for vehicular UE and pedestrian UE. Pedestrian UE with limited power capacities would only turn on their radio interface and perform PC5 transmission/reception during the on-duration of the DRX cycle. However, the use of DRX for pedestrian UE can result in longer delays or unreliable transmissions if not coordinated properly, especially when there are multiple PC5 communication sessions with different peers or groups. Different V2X services also have varying QoS requirements and traffic patterns, which need to be coordinated to maintain the QoS and power efficiency.

To support NR PC5 power efficiency for pedestrian UE, several issues need to be addressed. These include studying the applicability of NR PC5 DRX or other mechanisms to V2X operations for pedestrian UE, considering the impact on the V2X layer and the 5G Core Network (5GC). This involves studying how the 5GC can authorize and provision the DRX mechanism for pedestrian UE without degrading the QoS. Additionally, the coordination of requirements from different V2X services and determination of how the UE activates or deactivates the DRX mode over PC5 are important aspects to consider [75,76].

#### 3.1.6. 3GPP Release 18

Release 18 is marked as the start of 5G-Advanced. The evolution to 5G-Advanced with Release 18 was described as balanced, considering mobile broadband evolution, vertical domain expansion, market needs, and device/network evolution. Release 18 generated a list of topics for further discussion, including downlink MIMO evolution, uplink enhancements, mobility enhancements, additional topological improvements, XR enhancements, sidelink enhancements, RedCap evolution, NTN evolution, broadcast and multicast services evolution, positioning improvements, duplex operation evolution, AI/ML applications, network energy savings, and RAN enhancements.

From the V2X perspective, Release 18 continues to focus on enhancing sidelink communication to support vehicular devices in V2X (vehicle-to-everything) services. The objective is to improve the sidelink data rate by introducing carrier aggregation (CA) in sidelink communication, extending sidelink operation to the unlicensed spectrum, and enhancing sidelink support in the frequency range 2 (FR2). Additionally, the 3GPP is investigating mechanisms to enable LTE V2X and NR V2X devices to coexist in the same frequency channel, ensuring compatibility and an efficient utilization of resources. Furthermore, Release 18 aims to extend support for aerial vehicles, specifically unmanned aerial vehicles (UAVs), by introducing 5G NR capabilities. This is in continuation of the support provided for UAVs in 4G LTE, which included features such as enhanced measurement reports; the reporting of UAV height, location, and speed; flight path reporting; and signaling support for subscription-based aerial UE (user equipment) identification. These features will be incorporated into the 5G NR specifications as appropriate, enabling UAVs to leverage the benefits of 5G technology.

Overall, the 3GPP’s efforts in Release 18 aim to enhance sidelink communication for V2X services, support the coexistence of LTE V2X and NR V2X devices, and extend 5G NR capabilities to UAVs, catering to the evolving needs of various industries and facilitating seamless communication between vehicles and aerial vehicles [80,81,82].

#### 3.1.7. 3GPP Releases 19 and 20—Beyond 5G

The roadmap of 5G-Advanced (or beyond 5G) continues with Release 19 and Release 20 by the 3GPP. Nevertheless, in line with that, ETSI [77], ITU [67], and the 5GIA (5G Infrastructure Association) [118] are placing their recommendations as well.

The actual initial discussions on the concrete content of 3GPP Release 19 officially began in April 2023, with an anticipated completion of R19 specifications by December 2025 [78]. The actual 3GPP Work Items that are under discussion by the standardization working groups are listed in [79]. Current items include network sharing aspects, integrated sensing and communication, localized mobile metaverse services, studies of satellite access (phase 3) and UAV (phase 3), energy efficiency, the network of service robots with ambient intelligence, interworking with non-3GPP networks, multi-path relay, and more.

Such a detailed work-plan does not yet exist for Release 20—which is in the research phase. Hence, standardization details are non-existent. The same applies to Release 21—the first actual 3GPP 6G release—which is in its vision and requirements phases for 2023. Still, beyond 5G and 6G is an active research area, as they are expected to solve issues and cover demands already envisioned.

#### 3.1.8. Differences in the Communication Methods

As depicted in Table 2, the new modulation was the most significant improvement of Release 16. The most important difference between the C-V2X and DSRC technologies is the way they access the channel. The DSRC is based on WiFi technology. Thus, it uses CSMA, which implies that the transmitted frames are not aligned. The radio checks if a transmission is in progress, and if it is not, it tries to transmit the requested data. This a simple yet robust operation mode; however, it might not be optimal. Contrary to this, C-V2X technology uses radio resource blocks, which are assigned in a distributed way (in Mode 2 or Mode 4). The resource blocks require tight time scheduling, which is solved via GNSS signals (in Mode 2 or Mode 4) [61,65].

### 3.2. Security of 5G Networks: Secrecy Aspects of MIMO Antennas

The security of a telecommunication system has several aspects over the network layers, and one of them is the physical layer security (PLS), which discusses the radio channel properties. PLS is recognized to be an important field, with several possibilities and numerous available publications [55]. However, novel wireless methods are swiftly advancing, raising the need for the update of PLS. The antenna arrays used in modern telecommunication systems enable new security approaches and also possible threats, which are discussed in this chapter.

#### 3.2.1. MIMO Antennas and 5G Security

The antenna arrays used for Multiple-Input Multiple-Output (MIMO) applications has developed intensively in the last years. The field of telecommunication has changing requirements in capacity, latency, scalability, and reliability, which are the main keywords of modern 5G networks. To fulfill these requirements, amongst other solutions, antenna arrays were introduced to tackle the attenuation of the millimeter waves (mmW), introduce better data rates and capacities, and improve the channel stability. The number of antenna elements also increased rapidly in recent years, from 2×2 and 4×4 to 8×8 systems and beyond [63,73]. In terms of security, the most important utilities of antenna arrays include the beam-forming and channel estimation [56]. Multiple antennas in a system being fed in a specific way results in multiple beams or better single-beam properties that are scalable. Moreover, multiple antennas allow the characterization of the propagation channel, which can highly influence the data rate and channel capacity. These utilities raise security concerns that are different in nature from the usual PLS approaches; thus, they require additional attention:The channel properties are prone to the reliability of the beam-forming method; thus, the users have to be tracked dynamically.Multiple users at the same location can lead to interference and decayed channel properties.Channel state information (CSI) depends on the system of the antennas, which can be disturbed by external antennas.CSI estimation requires demanding computations as the size of the antenna array increases.

#### 3.2.2. Secrecy Models

The security of telecommunication systems is usually based on conventional cryptographic mechanisms, utilizing pre-shared keys [62,72]. Recently, the computational capabilities for deciphering these encryptions are swiftly improving, making malicious security attacks possible. Hence, physical layer security (PLS), using the physical system and propagation properties for increased security, has recently drawn attention and is being intensively researched [54]. The basis of this concept was developed by Wyner in 1975 [53] by introducing the wire-tap channel, i.e., the recently used model for physical security applications, including MIMO systems [51,70].

#### 3.2.3. Notations

Secrecy models are based on three main participants: one is the transmitter and another is the receiver whose initiative is to exchange information securely. However, there is a malicious participant, an eavesdropper, who aims for the information shared between the transmitter and receiver. This base conception is widely used in cryptography. Thus, for further simplification, the transmitter is often called Alice (A), the receiver Bob (B), and the eavesdropper Eve, which reduces the definition to information exchange between Alice and Bob, where Eve would like to gain insight or disturb the system [124,125], as shown in Figure 4.

#### 3.2.4. MIMO Channel Secrecy Rate and Capacity

The communication confidence between Alice and Bob can be quantified by the secrecy rate, which tells the rate of the information transfer to Bob and Eve. Ideally, Eve gets no information from the communication, which would result in a perfect secrecy rate. A specific arrangement of the telecommunication system between Alice, Bob, and Eve can be characterized by the secrecy capacity, which gives the maximum achievable secrecy rate [69,71]. Using this theory, an optimal transmission method can be chosen, e.g., transmitting independently over the MIMO subchannels for which Bob receives a better signal than the eavesdropper. In doing so, such a design can be created with the help of the available wire-tap channel models, which are closed-form calculations of the secrecy capacity [127,128].

### 3.3. Positioning Capabilities of 5G and Enablers

As assisted driving technologies are evolving and spreading, the requirements for their positioning capabilities are expanding: higher accuracy, a faster response time, and a more robust operation are needed, even under severe and special conditions. Unfortunately, currently, available positioning technologies cannot fulfill these increased expectations, not to mention the rising demands due to the increasing number of (semi-)autonomous vehicles.

The positioning technologies currently available to commercial vehicles can be categorized into three groups:Satellite-based positioning (GNSS);Base-station-based (2G, 3G, 4G);Hybrid solutions (DGPS).

The limitations of these technologies are known: they focus on the control problem or the situation analysis regarding localization, but they assume ideal operating conditions of the positioning systems [21].

The requirements for the positioning of driving-assisted vehicles are easily comparable to those in industrial applications, which are gathered in [16]. Moreover, there are well-defined overlapping test scenarios for the future challenges around positioning, among others:Valet parking using indoor positioning [35];Handling maneuvers in urban canyons (tunnels, dense cities);Asymmetric visibility scenarios [43].

There is a need to discover the possibilities of the application of improved localization provided by 5G to overcome these challenges.

Possibilities in 5G for localization are presented in [38], proving that it is worth trying to satisfy these requirements using 5G. It is worth noting that 5G does not have unlimited possibilities, as 5G is not used only for localization. Ref. [42] examined the trade-off between a millimeter-wave-based communication speed and localization precision in a multi-user scenario.

In this section, the currently used 5G positioning technologies are compared, and then the possibilities in hybrid solutions with 5G are put next to each other. Finally, cooperative localization techniques are examined, with further possibilities to improve 5G localization techniques.

#### 3.3.1. Localization Algorithms and Evaluation Methods Concerning 5G

As a radio-wave-based communication technology, 5G offers the possibility of using most positioning techniques and algorithms based on wave propagation. Ref. [43] gathers and compares six different possible positioning technologies. A survey on 5G positioning [50] also collects the basis of 5G-based localization and compares a few of the elementary methods. These methods are listed in Table 3. Choosing the right method depends on the specific application and is the designer’s responsibility. They have to set that the desired accuracy and refreshment rate targets and make a choice based on those requirements.

It is worth mentioning that, as 5G’s main application is communication rather than precise positioning, the performance of these methods can be worse in some areas than radio technologies specialized for localization (e.g., UWB). Despite this, the technological advancements in 5G ensure that even the stricter demands regarding positioning are met, and the spread of 5G offers a great economic advantage over other radio technologies.

#### 3.3.2. Hybrid Localization

An approach for improving localization within 5G is to use hybrid technologies. The results can be more accurate, the response time can be shorter, and the system is more robust, combining multiple diverse methods. In most cases, each component can independently provide localization information, so even if one part fails, the other can supply the system with position information during reduced operation.

Hybrid methods extend the ability of localization techniques to handle non-line-of-sight (NLOS) situations when the signal is diffracted or reflected, besides the ideal line-of-sight (LOS) situations. The improvement of regular methods using an intermediate approach with problem decomposition considering an end-to-end perspective is presented in [48].

The global navigation satellite system (GNSS) forms the basis of today’s technology for positioning, so it is an obvious choice to be one of the hybrid solutions’ resources. The possibility of combining this with other localization methods, like 5G-based or device-to-device (D2D) positioning techniques, is explored in [49].

A cellular information-based approach is presented in [44], with a physical-layer abstraction-based simulation methodology.

In [39], further information on the hybrid method sources is used to improve accuracy: the network density of 5G deployments and visibility mask of satellites.

Table 4 contains a brief summary of novel hybrid localization techniques using 5G. It is worth noting, however, since the technology of 5G is under development, the results are mostly provided based on a simulation-based evaluation. Nevertheless, it can be seen that there are great possibilities in hybrid techniques with forward-looking results that can be used as inspiration to improve these technologies further.

#### 3.3.3. Cooperative localization

In a significant proportion of cases, vehicles are surrounded by other vehicles (urban situations, highway traffic), so the possibility of cooperation appears to further improve the localization predictions. Ref. [33] summarizes the main aspects and advantages of cooperative localization. The paper lays the foundations for handling these methods and gives evaluation and comparison criteria.

One of the most important requirements for cooperative localization to work effectively is to have enough vehicles on the road using these technologies. As (semi-)autonomous vehicles are not widespread enough at the moment, the development and evaluation of these technologies are based on simulations rather than real-life data.

Because the development and research of these technologies are still in an early stage, current research papers propose different approaches and focus on one special aspect of this topic rather than providing a complete solution.

Ref. [47] presents one possible cooperative positioning and mapping method. It handles multi-path signals induced by the environment as a resource by creating a radio map and compares two different approaches. In [40], a different method is proposed, which can handle the challenges of the mapping and vehicle state estimation by using radio mapping based on large antenna arrays. The proposed algorithms exploit all available signal paths using a multi-model PHD (probability hypothesis density) filter and a map fusion routine.

Research regarding cooperative positioning is also present in other areas, e.g., industrial mobile networks, the results of which can also be utilized in vehicle-to-vehicle cooperative positioning. For example, seamless positioning using cooperation between mobile terminals (MTs) is presented in [46]. The paper investigates D2D communication in dense networks, examining centralized and decentralized structures using ranges and pseudo ranges. The Cramér–Rao lower bound is set to investigate the performance of these cooperative techniques, measured in a typical urban situation and environment.

It is also worth mentioning that cooperative positioning is also possible between different types of actors, like between vehicles and the mobile terminals of pedestrians, which can further improve localization performance.

Ref. [32] states that there are two main approaches in realizing these—possibly heterogeneous—cooperative localization systems: centralized and distributed implementation. The advantages and disadvantages of these approaches are compared in [32], using array processing based on GNSS-only information and heterogeneous information.

## 4. 5G and V2X Deployment Projects

V2X projects and consortia bring together automakers, network operators, government agencies, and other stakeholders to collaborate on the development and deployment of C-V2X technology. By pooling resources and expertise, these groups can accelerate the pace of innovation and help to overcome the challenges of deploying C-V2X on a large scale. This section aims to provide an overview of recent C-V2X deployment projects and their use cases. We will examine the challenges and opportunities of deploying C-V2X and explore the ways in which the technology can benefit from these projects.

Ref. [7] summarizes the main communication activities, milestones, enabling technologies, and standards, mentioning 5G-CARMEN and 5G-DRIVE. However, it does not mention the exact impact and results of such projects and consortia. This section discusses these results and several others in more detail.

### 4.1. 5GCAR

The 5GCAR project was initiated in 2019 at the UTAC-TEQMO test track in France. It implemented a 5G-based communication system to address various automotive use cases, with the core of the system comprising 5G new radio sideline communication, 5G new radio positioning, and cellular communication using 5G deployment models like edge computing and network slicing [129].

To optimize the merging process for vehicles entering a lane, 5GCAR utilizes an IoT platform that includes a maneuver planning system. This system communicates with connected vehicles by sending information from a central maneuver planning system. It also collects data from roadside cameras to provide real-time video between nearby vehicles through vehicle-to-vehicle (V2V) communication, facilitating safe overtaking maneuvers. Connected vehicles within the 5GCAR system transmit collected Lidar information, which estimates the position and speed of detected vehicles, to the vehicle-to-infrastructure (V2I) server infrastructure in the IoT platform. The server analyzes these data to predict the real-time trajectories of the vehicles and estimate the risk of collision. It then provides warnings to affected drivers to safely and comfortably avoid potential collisions [130].

Another significant aspect of 5GCAR is its road user protection system. This system is designed to detect and identify pedestrians crossing the street behind obstacles. It can halt the vehicle when a pedestrian is in close proximity and issue collision warnings to the driver in advance [131].

### 4.2. 5GCroCo

Fifth-Generation Cross-border Control (5GCroCo) is one of the related projects and activities in the EU context for validating 5G technologies for cooperative, connected, and automated mobility (CCAM) in the cross-border corridor along France, Germany, and Luxembourg to implement large-scale cross-border scenarios. Moreover, the project is also aiming to define unprecedented business models integrated with 5G features, such as network slicing, new radio, MEC-enabled distributed computing, predictive QoS, and improved positioning systems made up of reliable connectivity and combined services to enable innovative use cases for CCAM [88].

The project 5GCroCo launched on 1st November 2018 with a focus on the identification of the requirements for CCAM in cross-border scenarios and contributions to future releases of communication standards in addition to locating and defining novel business models that can be built on top of new CCAM services, which are facilitated by 5G technologies along the cross-border project deployment [85].

### 4.3. 5G-DRIVE

A project called 5G HarmoniseD Research and TrIals for serVice Evolution between the EU and China (5G-DRIVE) launched in 2018 and involved 17 European partners from 11 countries. It started with the first draft for a generic trial plan carried out in two main pilot sites in the EU: Espoo in Finland and JRC at Ispra in Italy. The project is a transparent business model that deploys 5G technologies and benefits on pre-commercial test-bed V2X services and, after that, demonstrates internet of vehicles (IoV) services using V2V and V2N communication scenarios.

The 5G-DRIVE project aimed to achieve specific objectives of the integration between 5G and IoV by deploying tests and use cases on 5G-based IoV scenarios, such as assuring interoperability between European and Chinese IoV technologies, validating 5G enhancements and features, such as bandwidth and latency, and evaluating V2V and V2N communications under real-life conditions, such as security attacks and interference. The 5G-DRIVE project is focusing on EU–China (Chinese twinning project) collaboration development based on 5G extensive dissemination and innovative V2X solutions [132].

### 4.4. 5G-CARMEN

The 5G-CARMEN project was launched on the first of November 2018 for deploying 5G solutions on the critical corridor along 600 km of roads, crossing Bavaria in Germany, Tirol in Austria, and Trentino/South Tyrol in Italy, connecting three European regions, which are also known by Munich–Bologna as a 5G corridor for connected vehicles. The 5G-CARMEN project aimed to provide a multi-tenant platform for the automotive sector to make mobility smarter, greener, and more intelligent, focusing on the ultimate goal of enabling self-driving vehicles.

The 5G-CARMEN project consists of V2V, V2I, and V2N communications and investigates four 5G V2X use cases and application scenarios. The first use case is cooperative maneuvering by providing a platform enabling vehicles to exchange trajectories/maneuvers and determine optimal behavior to avoid dangerous situations and positions, using speeds and other helpful data. The second use case in 5G-CARMEN is extended situation awareness using V2I communication to obtain information about the surrounding environment and detect potentially abnormal situations. Video streaming is the third use case in 5G-CARMEN, which is exploring different network architectures and configurations to achieve permanently available high-quality video streaming, including in cross-border scenarios. The fourth use case is green driving by promoting greener driving styles due to providing intelligent solutions, such as supporting road operators to determine decisions to help limit the negative impact of transportation on the environment and promote the approach of hybrid vehicles [89].

### 4.5. 5G-MOBIX

The 5G-MOBIX initiative, an EU-funded project conducted from 2018 to 2022, brought together 60 partners from 13 countries to develop and evaluate automated vehicle functionalities using 5G technology. The project focused on two cross-border corridors and six urban trial sites, aiming to assess the benefits of 5G for connected and automated mobility (CAM) applications. The trials conducted as part of 5G-MOBIX allowed for the assessment of various CAM applications, including cooperative overtake, highway lane merging, truck platooning, remote driving, and vehicle quality of service support. These applications were tested in real cross-border environments, utilizing multi-access edge computing (MEC) infrastructures and exploring the interworking of multi-operator deployments. The project aimed to gain technical insights and identify new markets for CAM applications and edge service developers.

The trials considered diverse traffic; network coverage; service demand conditions; and legal, business, and social aspects at each site. They evaluated the performance of different protocols, scenarios, and 5G features, addressing challenges related to cross-border functionality, telecommunication infrastructure, application security, privacy, and regulations. The project took a holistic approach to evaluation, considering technical aspects, impact factors, and alternative approaches to solutions. The deployment of trial sites in 5G-MOBIX provided the opportunity to test various configurations; assess interoperability between systems and components; and gain insights, lessons learned, and best practices. The results and specifications from these trials significantly contributed to the research and development of 5G for CAM. Mobile operators benefited from cooperation with neighboring operators and road operators, gaining valuable insights for future trials and enhancing their understanding of limitations, roaming, and cross-network interconnection.

The project’s findings on 5G performance in connected and automated vehicles (CAVs) operations offered a realistic assessment of technology capabilities and demands. This information, along with the specifications of the 5G-augmented vehicle and roadside infrastructure architecture, guided further development and deployment in the automotive industry and transport sector. In addition to technical aspects, 5G-MOBIX also examined standards, regulations, and cost considerations associated with 5G roll-out. The analysis highlighted gaps in standardization, policies related to autonomous vehicles, infrastructure requirements, roaming charges, data privacy, security, and the costs associated with 5G deployment. The project aimed to bridge these gaps and provide valuable insights for stakeholders involved in the development and implementation of 5G for CAM services and associated business models [90].

### 4.6. 5G-ROUTES

The main objective of the 5G-ROUTES project is to conduct advanced field trials for innovative Cooperative Awareness Message (CAM) applications along the designated 5G cross-border corridor known as “Via Baltica-North”, which spans across Latvia, Estonia, and Finland, aiming to validate the latest 5G features and 3GPP specifications under real-world conditions. The project aims to accelerate the deployment of 5G end-to-end interoperable CAM ecosystems and services in digitized motorways, railways, and shipways throughout Europe.

The project is built upon three pillars. Firstly, it focuses on formulating and validating a wide range of innovative use cases that directly relate to CAM and cross-border mobility in the “Via Baltica-North” corridor. These use cases align with 3GPP V2X Phase 3 and will undergo validation against more than 150 target 5G Key Performance Indicators (KPIs). Secondly, the project integrates various enabling technologies to facilitate the execution of field trials. These technologies include AI-based network slicing and optimization, AI-based distributed mobile edge computing (MEC) for CAM services, AI-based 5G radio interface for shared spectrum usage, AI-based positioning enhancements for vehicle-to-everything (V2X) communication, and cross-domain integration fabric for multi-domain interaction and service roaming, as well as complementary technologies, like satellite 5G connectivity, vehicular On-Board Units (OBUs), IEEE 802.11p, and the FRMCS (Future Railway Mobile Communication System).

Lastly, the project leverages a Key Performance Indicator (KPI) visualization system from the 5G-SOLUTIONS project, extending its functionality to incorporate CAM use cases. This system enables the parallel execution of use cases, automated analysis of trial results, and near-real-time presentation through an intuitive, user-friendly dashboard. The project also aligns its implementation roadmap with the latest 3GPP standardization releases, actively contributing to standardization bodies and promoting a collaborative approach between the telecom and automotive sectors. The project’s objectives follow the SMART criteria, including developing innovative CAM use cases, analyzing requirements, advancing enabling technologies, developing infrastructure, conducting field trials, managing exploitation and innovation, contributing to standardization, and ensuring long-term success through dissemination and collaboration.

Overall, the 5G-ROUTES project aims to validate and optimize 5G technologies for CAM services in a cross-border context, with the ultimate goal of driving the widespread adoption of interoperable CAM ecosystems and services in the transportation sectors across Europe [92].

### 4.7. 5G-Blueprint

The 5G cellular V2X aspect of the 5G-Blueprint project focuses on designing and validating technical architecture, business models, and governance models for cross-border teleoperated transport based on 5G connectivity. The project aims to explore the economic, governance, tactical, and operational aspects of implementing 5G-enabled transport and logistics solutions. The goal is to increase efficiency, reduce manpower shortages, and improve safety through teleoperation, data sharing, and AI utilization. The project involves several use cases, including automated barge control, automated docking, CACC-based platooning, and remote takeover. These use cases demonstrate the potential of 5G connectivity for various aspects of transport and logistics, such as improving port entry efficiency, optimizing docking operations, and enabling the remote control of vehicles and vessels.

To achieve these objectives, the project sets technological, business, and governance goals. Technological objectives include designing and implementing a 5G network with a low-latency, reliable connectivity, and high throughput, as well as developing and deploying a teleoperated system and ensuring its safety. Business objectives involve market analysis, identifying commercial possibilities, promoting teleoperated transport, and facilitating the adoption of 5G-based solutions. Governance objectives focus on identifying and addressing regulatory issues related to cross-border teleoperated transport based on 5G connectivity. The project also addresses specific challenges and considerations related to 5G technology. It explores the combination of URLLC and eMBB to support teleoperation with a high bandwidth, a low latency, and reliability. Additionally, it validates the use of 5G in challenging environments, such as port terminals and critical infrastructure, and examines concepts like network awareness and multiple slices on the same user equipment. The project leverages 5G’s capabilities, including network slicing and multi-access edge computing (MEC), to optimize connectivity and performance for different applications and scenarios.

The outcome of the project will be a blueprint for the operational deployment of teleoperated transport solutions in the logistics sector and beyond. It aims to pave the way for the widespread adoption of 5G-enabled teleoperation and enhance cross-border transport efficiency and safety [93].

### 4.8. 5G-MED

The 5GMED project, funded by the ICT-53-2020 program, aims to design a common 5G infrastructure architecture for roads and railways. The project focuses on delivering sustainable business models, investment viability, and scalability for connected and automated mobility (CCAM) and Future Railway Mobile Communication System (FRMCS) services across European borders. The project has several technology objectives, including specifying a scalable and cross-border 5G and AI-enabled system architecture, designing cross-operator service orchestration, establishing cooperation models between stakeholders, and identifying priorities for FRMCS performance. The project also aims to contribute to standardization activities, perform cost/benefit analysis, define innovative business models, promote the impact of outcomes, and ensure scalability and replication. The testing facilities located in Spain and France will be used to validate the latest 5G technologies and architectures in cross-border CCAM scenarios. The project also focuses on characterizing 5G for advanced CCAM use cases, validating sustainable models combining 5G and AI, supporting railway use cases, and developing a pan-European cloud infrastructure. Ultimately, 5GMED aims to leverage public and private investment in 5G deployment to make CCAM a reality in cross-border corridors in Europe [94].

### 4.9. VITAL-5G

The VITAL-5G project aims to establish an open, virtualized, and flexible experimentation facility for the Transport and Logistics (T and L) sector, utilizing 5G connectivity. Its main focus is on developing, managing, and sharing innovative network applications specific to the T and L industry. The consortium recognizes several key challenges in the adoption of 5G for production services in the T and L sector, including the network-centric vision, the mismatched backgrounds between telecommunication companies and verticals, and the limited availability of vertical-specific network applications. To achieve its objectives, the project has set strategic goals. Firstly, it aims to develop and share novel vertical-specific and vertical-agnostic network applications tailored to the T and L sector. This involves supporting open-source tools and an open repository to facilitate the development and sharing of these applications. Secondly, the project aims to deliver an open, virtualized 5G-enabled testing and validation experimentation facility. This facility will provide a platform for relevant T and L stakeholders to deploy and benchmark their innovative network applications on a 5G network.

Additionally, VITAL-5G aims to demonstrate the benefits of 5G connectivity for advanced multi-modal logistics services. By showcasing the added value of 5G in creating a functional value chain for highly automated freight transportation across European roads, seas, and rivers, the project aims to highlight the advantages of 5G in the T and L sector. Furthermore, the project aims to provide customized and virtualized access to network and T and L infrastructure. This will enable dynamic and tailor-made service provisioning to third parties, such as SMEs, allowing them to validate their applications over otherwise unavailable resources and boosting their confidence prior to actual deployment.

VITAL-5G also focuses on the development of novel business models for integrated and cooperative services. By facilitating the development of open and cooperative services across multiple domains, the project aims to address specific T and L use cases and justify investments from key stakeholders. Moreover, the project aims to advance a T and L-centered ecosystem by driving the integration of 5G services into the T and L vertical. This involves fostering collaboration between key stakeholders, such as port authorities, road operators, mobile network operators, and SMEs developing cutting-edge technology and applications. To achieve these goals, VITAL-5G follows a methodology that includes activities such as developing a functional architecture, onboarding and upgrading existing 5G testbeds, designing T&L-related network applications, and conducting extensive technical and business validations through real-life trials. The project emphasizes collaboration with ongoing 5G-PPP projects and seeks to contribute to standardization organizations while disseminating its results widely [95].

### 4.10. Motorway Measurement Campaign for Automated Driving Technologies in Hungary

A measurement campaign was carried out on a real-world motorway stretch of Hungary with the participation of international industrial and academic partners. The measurement campaign had a dual purpose. Firstly, it involved mapping the road geometry with exceptional accuracy to create an ultra-high-definition (UHD) map of the test road. Secondly, the vehicles, equipped with differential global navigation satellite systems (GNSSs) for precise ground truth localization, conducted specific test scenarios while gathering comprehensive data using various sensors. Throughout the campaign, both the vehicles and the infrastructure recorded all the test runs [84].

The measurement campaign was carried out on a highway section near the town of Csorna, in the north–western part of Hungary (Győr-Moson-Sopron county, Western Transdanubia Region). Csorna is located at the crossing of two main regional highway sections, one is the M85 running east–west, connecting the Fertő-Hanság region to Győr and via M1 to the Capital City of Budapest and the other way to Burgenland in Austria and via A3 to Vienna. The other one is the M86 running from north to south, which is part of the TEN-T comprehensive network, and part of the E65 route running from Sweden (Malmö) to Greece (Crete) at the same time. Both of the roads are part of the Hungarian State Public Road Network, and they are managed by the national road operator company, Hungarian Public Roads.

The main lessons and contributions of the measurement campaign are as follows:Planning and managing a measurement campaign with several partners and with different sensors is a huge and complex task where success relies on thorough preparation.For high-precision mapping, two datasets were collected using different high-tech instruments during the measurement campaign. Different capabilities of sensors are needed when identifying small details, such as supplemental traffic signs, or when creating a surface model of the ground.Ground truth information for object detection algorithms is of crucial importance in the automotive testing field. The acquired point clouds and image recordings combined with the shared ground truth position information can be directly used for testing and validating neural network-based object detection algorithms.The presented two application examples demonstrate the viability of the collected data during the M86 Measurement Campaign. This dataset may support a large variety of solutions for the testing and validation of different kinds of approaches and techniques.Fifth-generation network tests were carried out under different radio conditions. Different measurement scenarios provided latency results that behaved as expected beforehand.

### 4.11. Central System for Supporting Automated Vehicle Testing and Operation in Hungary and Austria

The Central System project is focusing on realizing a transport system using the highest level state of the art technology to develop and demonstrate a holistic solution to support and operate autonomous vehicles in cooperation with infrastructure elements. The project targets the implementation of the system for testing purposes in the beginning but should evolve for transport operation and control. In today’s mainstream, for the operation of highly automated vehicles, the main environmental model for vehicle guidance is generated based on the ego vehicle’s perception system and extending the own local sensor signals with infrastructure and other vehicle’s information [91].

The proposed solution of the project builds up the global environment model externally in a cloud and supports the individual vehicles with a comprehensive world model and can even control them. The system will collect all information from both the vehicle and infrastructure side and fuse them together in a cloud-based, real-time digital twin. The cloud-based map content includes the following:A static map (3D representation, vectorized data, material properties);Semi-static and semi-dynamic content (like weather or lighting conditions, road conditions);Full dynamic data (vehicles, pedestrians and all relevant dynamic information).

The system properties include the following:A real-time fusion of data using vehicle and infrastructure sensor data;An ability to record data;The use of applicable international standards;An ability to support vehicles with a real-time environment model;A system prepared for controlling infrastructure elements (e.g., traffic lights) or even vehicles;Support for future testing procedures for connected and automated vehicles (CAVs);Scalable, reusable, and future-proof architecture.

The projected outcomes of the campaign include a real-time digital twin service that can be used for supporting test track or public road-based connected vehicle functionalities, for example, for cooperative perception-based functionalities. A low-latency and high-bandwidth special 5G network service is applied for communication. Relevant traffic situations are detected automatically and recorded for scenario-based testing, which can be used in CAV development. Besides that, a real-time mixed reality testing service for CAV testing or special testing, like vehicle in the loop and scenario in the loop methodologies, can significantly reduce test costs and reduce the development time of vehicles. The testing is mainly applicable on proving grounds; however, testing on public roads enhances the proof of functionality.

The detailed objectives of the project regarding 5G communication are as follows:Examine and describe the connection between UE terminals and related systems. The project presented novel scenario-in-the-loop measurements and results, where a 5G communication link provided a stable, real-time connection between the real world and its virtual representation. This means that the vehicle under testing does exactly the same thing in reality and in parallel real-time in the virtual space. It responds real-time either to real obstacles on the test track or obstacles that are generated in the virtual space, such as the dummy during the demo.Analyze and evaluate the configuration on the 5G data network—in terms of 5G radio, IP transmission technology, and the 5G Core. A key objective of the measurement campaign is to monitor the performance not only with end-user metrics but on the mobile network side using RAN, IP, and Core statistics with the help of the mobile network operator.Present a life cycle model specializing in 5G data transfer technology and data transfer protocols. The life cycle model should define how a cellular V2X device should be managed during the operation of the device. The model should describe how the operator should introduce new network features in case of rollout and how malfunctions should be handled.Develop a measurement procedure for the low-latency services for V2X. The project aims to present real-life 5G-based V2X communication scenarios with actual network traffic measurement results, including presents latency, round-trip time, and packet interarrival time results in these real-life scenarios under 5G architecture.Create an M2M prototype product for V2X communication for fixed and mobile devices.

### 4.12. Quality-on-Demand API for Automated Valet Parking

Deutsche Telekom, in partnership with the Fraunhofer Institute for Material Flow and Logistics (IML) and BMW, has successfully demonstrated automated valet parking using 5G technology. The trial was conducted at the Hamburg Airport in Germany, where a vehicle equipped with automated driving technology and a 5G connection was able to park itself in a designated parking space.

Automated valet parking has the potential to revolutionize the parking industry, reducing the need for parking infrastructure and improving the overall driving experience. Drivers would no longer need to spend time searching for parking spaces or maneuvering in and out of tight spaces, as the vehicle would be able to park itself. This would also reduce the risk of accidents and increase the efficiency of parking operations.

The successful demonstration of automated valet parking using 5G technology also highlights the potential for other applications of connected and automated driving technology. For example, vehicles could communicate with each other and with their surroundings, improving road safety and reducing congestion. Additionally, 5G connectivity could enable new services and business models, such as in-car entertainment and e-commerce [96].

### 4.13. Real-Time Racing Experience on 5G Network

HUMDA Hungarian Automobile-Motorsport organized a campaign where the two touring cars zigzagging around the Hungaroring asphalt track could be raced sitting in the simulator in the pit lane building. The digital twin pilot project used the blended reality function to display events on the simulator screen in real-time. Both cars were equipped with a pilot device that utilized positional, diagnostic data and a live video stream via a 5G Non-Public Network (NPN).

To ensure accurate positioning and data collection, a data processing system and data package management tool were developed. The collaboration among participating organizations was crucial for the development of visualization software and algorithms. Sensor data collected by the vehicles were rapidly transmitted via the 5G NPN network to the simulation environment and to a server for analysis. The 5G cells deployed along the race track ensured reliable transmission with minimal delays [97].

### 4.14. Live Trial of 5G-Connected Car Concept To Launch in Turin, Italy

On 2 December 2021, a live connected car trial took place in Turin, Italy, showcasing new roaming technology for traffic safety. It was organized by the 5G Automotive Association (5GAA) in collaboration with eight member companies representing leading technology firms from around the world. The objective was to demonstrate the near-real-time notification of roadway hazards through 5G edge networks, enabling improved driver and pedestrian safety. The trial highlighted the potential of 5G and edge technology in delivering connected services that can transform daily life and contribute to the digital transformation of smart cities. With high-speed 5G transmission and edge servers equipped with AI capabilities, smart city technologies, such as real-time traffic management, became feasible. The connected car concept leveraged this technology to communicate traffic hazards to drivers and pedestrians via user-authorized mobile apps, facilitating safer and more efficient navigation.

The live trial specifically focused on addressing the technical challenge of enabling seamless communication between different network operators as vehicles cross borders. Roaming services, which allow uninterrupted communication regardless of the network, were successfully demonstrated. This trial was the first of its kind in Europe and aimed to prove the effectiveness of the connected car concept in a roaming scenario. The trial involved multiple objectives, including testing interworking between different network operators, ensuring global operational availability for OEMs, and seamless V2X service transfer in a multi-network operator roaming scenario. The collaborative effort involved a large ecosystem of stakeholders, including car manufacturers, mobile network operators, and technology providers, highlighting the importance of public–private sector cooperation in adopting and advancing 5G-connected car technology.

The trial provided valuable insights and learnings for various players in the industry, paving the way for future development of 5G-connected car services at the edge. The City of Turin, along with its Smart Road infrastructure and technical support from 5T, played a significant role in facilitating the trial. The success of the trial showcased the transformative potential of 5G and edge technology in revolutionizing the automotive industry and creating safer and more connected transportation systems [98].

### 4.15. Live Trial of 5G-Connected Car Concept Launches in Blacksburg, Virginia (VA)

On 24 March 2022, the Commonwealth of Virginia hosted a live trial of a groundbreaking driver and pedestrian safety concept utilizing 5G and edge technologies. This trial, organized by the 5G Automotive Association (5GAA) in collaboration with eight member companies representing leading technology firms worldwide, aimed to enable near-real-time notification of roadway hazards. It was one of three international trials conducted as part of a historic public–private collaboration, with a related trial taking place in Turin, Italy. However, the North American live trial held in Blacksburg, VA, on the Virginia Smart Road operated by the Virginia Tech Transportation Institute, marked a significant milestone as the first of its kind in North America.

The primary objective of the trial was to demonstrate how connected mobility and the use of 5G and edge computing technologies could enhance safety on roads. By leveraging the high-speed transmission capabilities of 5G networks and the computing power of edge servers, smart city technologies, like near real-time traffic management and various business applications, could be enabled. The connected car concept utilized this advanced technology to establish communication between car sensors, pedestrian smartphones, and a user-authorized mobile app. Through this communication, crucial information regarding traffic hazards, such as accidents and road construction, could be shared, ensuring both pedestrian and driver safety while facilitating efficient navigation. One of the key technical challenges addressed by this international trial was the seamless communication between different communication service providers (CoSPs) managing the 5G networks. Each CoSP operates according to geographical regions, and their edge solutions must be capable of uninterrupted communication for V2X (vehicle-to-everything) applications as drivers cross borders. The successful demonstration of roaming services, which allow uninterrupted communication regardless of the network, served as an early achievement for multi-mobile network operators (MNOs). The core objective of the live trial was to showcase that the connected car concept could work seamlessly in a roaming scenario, representing the first attempt of its kind in the United States. The trial aimed to serve as a stepping stone towards realizing the potential for automakers to provide services to their connected cars through the network.

The Virginia live trial focused on three main objectives. The first objective was to address the multi-mobile network operator scenario, examining how a vehicle connected to one MNO could use an MEC application operated by another MNO without losing the benefits of low latency. The second objective aimed at ensuring global operational availability, exploring how an original equipment manufacturer (OEM) developing a MEC application could ensure its consistent functionality and performance across different MNOs on a global scale. Lastly, the trial aimed to demonstrate a multi-MNO scenario with roaming, enabling a seamless transfer of the V2X service from one operator to another as the car OEM moves across geographic locations in a roaming scenario. Apart from the technical aspects, the trial emphasized the value of collaboration within a large ecosystem to develop the technology and build a compelling business case. This collaboration was crucial in driving potential market adoption of 5G and C-V2X (cellular vehicle-to-everything) technologies, contributing to the digital transformation of smart cities in the future. The involvement of both the public and private sectors was necessary to adopt this new technology and provide a better quality of life for all citizens [99].

Table 5 summarizes the introduced projects, while Figure 5 shows which project focuses on which 3GPP V2X use case scenarios along with the end date of the projects. The figure highlights the fact that the extended sensors and remote driving use cases are the most researched ones in these projects.

## 5. Obstacles and Challenges in Evolving V2X Communication

### 5.1. RAN Obstacles and Challenges

#### 5.1.1. The Used Frequencies

Due to the cooperative property of the V2X network, it would be essential to use the same access layer technology in all vehicles. The C-V2X and the ITS-G5 technologies target the same 5.9 GHz frequency range, which is divided into channels. This frequency range is dedicated to ITS communication. The lack of backward compatibility of Release 16 can also be a problem since the parallel usage of Releases 14/15 and Release 16 would require even more bandwidth. There are concerns about the parallel use of these technologies on subsequent channels as well [133].

#### 5.1.2. Subcarriers or CSMA/CA—The Perks and Drawbacks

As shown in Table 2, 5G (and 4G) uses semi-persistent sensing of the least-occupied resource technique, contrary to the 802.11 family’s CSMA/CA. The latter technique divides the available frequency band into so-called radio resource blocks. The allocation of these resource blocks is performed distributively. This technique has pros and cons as well.

One of the key advantages is the enhanced range. The reason for the enhanced range is that C-V2X uses relatively higher transmission power. The 5.9 GHz band is allocated for ITS safety purposes, and the communication on this channel is essentially distributed. Thus, the law specifies the maximum allowed transmission power for the devices in order to avoid congestion on the channel. The difference between the DSRC and the C-V2X is that the latter uses a tighter frequency band with the same transmission power to transmit the message. The tighter band results in less noise, eventually resulting in a higher range.

The other perk of the usage of radio resource blocks is that the channel allocation can be performed by an external entity, which increases the channel utilization. However, this method would introduce a single point of failure, which is not desired in such systems [134].

This approach has some drawbacks as well. The tight resource block alignment introduces a dependency on GNSS signals (in Modes 2/4), which can unnecessarily limit communication in urban canyons or tunnels. The simultaneous transmission of the packets can increase the probability of simultaneous transmission, which results in packet loss. There is a little additional latency added to the system as well due to the scheduling. The fixed radio resource blocks can result in degraded utilization in the case of varying packet lengths, which is typical in standardized direct V2X communication (like CAM messages).

As we can see, the modulation of these communication technologies is relatively similar. However, the channel access (the MAC) design has several implications for the performance of these communication technologies. Figure 6 from [135] highlights the differences relatively well. The DSRC performs slightly better in the case of low distances; however, the C-V2X outperforms the other technology in the case of higher distances.

### 5.2. Security Risks of 5G RAN

There are two approaches to eavesdropping on a communication system: passively and actively. Passive eavesdropping will not modify the original communication system but gain information from it. However, active eavesdropping will modify the system itself in a way that information can be obtained easier. The available publications discuss several scenarios of eavesdropping and the possible steps to avoid it.

#### 5.2.1. Passive Eavesdropping

For passively gaining information from the communication system, the multiple antenna systems proved to be efficient for prevention. In theory, if the number of antennas is approaching infinity (i.e., a large number), then the secrecy of the channel is almost perfect [111]. This property is based on the beam-forming of the MIMO system, which makes it possible to focus on Bob and avoid Eve. The first, and, so far, only, experimental evaluation of passive eavesdropping in a massive MIMO system is discussed in [107].

As it seems, a system with an MIMO base station (BS) is not prone to passive eavesdropping, which is a huge security development from the single antenna systems. However, measurements published recently showed that there are deviations between the theory, using the Rayleigh channel model and the real scenario, as the number of antennas is increasing in the MIMO system, giving Eve a higher channel capacity than expected. Also, in a realistic scenario, Eve can change position and find the optimal place for the best channel capacity (Figure 7), hence, increasing the chance of acquiring information. Furthermore, an MIMO antenna for Eve would result in further passive attack possibilities [110]. Thus, the developing techniques open up possible security gaps, and novel solutions are required.

#### 5.2.2. Pilot Contamination (PC)

The estimation of the propagation channel properties in telecommunication radio systems is performed by pilot transmissions. The channel state information (CSI) must be estimated frequently for MIMO systems because of the large number of antennas and the relatively short channel coherence time. Thus, this system is sensitive to incorrect CSI estimation, which degrades the communication link properties. Hence, there are already existing threats called pilot contamination (PC) attacks, where the malicious participant is broadcasting equivalent pilot signals, making Alice use a false CSI estimation. As the system is expecting an altered channel rather than reality, the transmission will perform poorly. Alice will use a non-optimal beam-forming and power levels, which would result in a possibility for increased vulnerability to passive attacks [104]. The detection of such threats is already studied [101], but the secrecy aspects are not yet available. Recent publications propose possible counter-measures, e.g., keeping the pilot assignments hidden and also using a pilot set that scales with the number of antennas in the system [104]. Furthermore, a possible solution can be the use of a smart algorithm (machine learning) that can detect PC attacks and alter the pilot assignment and the CSI estimation system [105].

#### 5.2.3. Modulation and Coding Scheme (MCS) Saturation

As a consequence of the adaptive properties of the communication link, the channel stability can be maintained in case of a poor channel, e.g., by applying a lower degree of modulation. However, if the channel is sufficiently strong, then the level of modulation can be higher, providing better channel capacity. This adaptive approach alters between discreet levels of modulation and coding schemes (MCSs). The set of a proper MCS is also essential for a secure MIMO communication, as usually Bob has a higher signal-to-noise ratio (SNR) than Eve (beam focusing), and a high enough MCS selection permits Bob to receive but not Eve with the lower SNR. Hence, a problem may arise when the channel properties are strong enough but the system already reached the highest MCS. Thus, Eve can have a high enough SNR to receive even the packages with the highest MCS, as Bob will not increase it further. In summary, the proper MCS selection can provide security against eavesdropping but also has limitations. MCS saturation can happen systematically, as a result of specific system features, and also artificially, e.g., by altering the channel measurement results (pilot contamination). Hence, to avoid MCS saturation flaws, the communication system has to contain adequate power control and further security aspects preventing false channel measurements [107,110].

#### 5.2.4. Active Uplink Jamming (Pilot Spoofing)

The CSI of the given link between Alice and Bob has to be estimated frequently and can be the source of security issues, as discussed with the pilot contamination attack. Similarly, Eve can transmit the same pilot sequence as Bob, but Eve can also use this method to make Alice confuse Eve with Bob. Hence, Eve can take over and alter Alice’s beam-forming and MCS decisions in a way that means Eve has better channel properties toward Alice than Bob has. As a consequence, Alice will consider Eve a legitimate user. In this scenario, called pilot spoofing, Eve can control the leakage information to receive by controlling its active training power [108]. In [106], it is proved that if the number of antenna elements or the length of the training vector is approaching infinity (i.e., a large number), then this kind of attack is ineffective. Similarly, the same encryption methods can be applied, as with pilot contamination, to increase security. Another approach against an active jammer is allocating the antennas of the communication system into two groups: one is used for the information exchange, and one is used for identifying and jamming the eavesdroppers [102].

### 5.3. Further Physical Layer Security Approaches

Apart from the widely known methods tackling physical security issues of MIMO systems, there are also novel and alternative approaches.

#### 5.3.1. Leaky Wave Antenna (LWA)

The antenna structure does not necessarily have to be an array to have physically secure behavior. The so-called leaky wave antennas can also be used for this purpose and have angle-dependent frequency characteristics [100]. Thus, beam-forming happens via the carrier frequency change, which gives an additional level of security, having not only spatial beam-forming but carrier frequency differences between separate users (Bob, Eve). In doing so, the secrecy capacity is forced to be high when Bob and Eve are at different angles. However, when the angles of Bob and Eve are approaching, then Eve can gain a higher SNR than Bob, which would result in low secrecy capacity. Moreover, for the edge frequencies (max. and min. angles), the secrecy capacity is lower for small angle differences between Bob and Eve. Although this phenomenon also appears with conventional antenna arrays, in the case of the LWA this less secure edge behavior applies for significantly different angles of Eve, which makes this system more resilient to single Eve attacks. Hence, such antennas have great potential in realizing a secure transmission due to these unique properties [136].

#### 5.3.2. Physical Layer Key Generation

The secrecy of the communication system can be provided through an adequate encryption method, which is complicated enough to decipher. However, such methods require demanding computational resources that are not available in the case of small-sized, mobile devices, as they are usually designed for low energy consumption as well (e.g., internet of things or wearable devices). These applications require alternative methods to create a secret key for communication secrecy. One of these methods is the physical key generation, where the key is calculated from the individual and mutual movement trajectories of the devices, which is measured via the received signal strength (RSS). This method assumes that specific devices have similar movement properties, which makes them unique, unlike the possible eavesdropper [103].

Physical layer security applications have drawn attention recently due to their low computational demands and applicability for modern wireless multi-antenna systems. There are several secrecy flaws that are observed and well studied. However, for the applicability to the new MIMO and 5G systems, these methods require modifications or even the introduction of novel ones, as such methods are still scarce. After the overview of the recent and perspective sight methods, it seems that several aspects are not studied yet, requiring further research [109]:The effect of the eavesdroppers’ mobility;Multi-user MIMO scenarios;Backscatter communication perspectives;An eavesdropper with an MIMO antenna [110].

As it seems, published methods for providing physical security have considered only a section of possible threads, raising the importance of investigating new scenarios, e.g., a system of several participants and eavesdroppers using MIMO antennas, and advanced algorithms, i.e., artificial intelligence, to cause malfunction or gain information. The summary of the introduced risks are presented in Table 6.

### 5.4. Use Case Specific Challenges

#### 5.4.1. Communication Range

One of the most significant perks of PC5 (and Uu)-based communication over 802.11p-based communication is the enhanced communication range. However, the increased range means an increased number of received packets to handle as well. A higher number of received packets means that processing the packets also requires more effort, and, practically, a more powerful—and more expensive—processor unit, which leads to higher equipment costs. This means that the higher range also requires a strong use-case-based justification. If we calculate with two vehicles approaching each other with a 300 km/h relative speed difference, a 1 km range results in a 12-s time for scenario mitigation in the case of unchanged speeds [133].

#### 5.4.2. Vulnerable Road Users

Vulnerable road user (pedestrians, cyclists, etc.) use cases are active research areas in the V2X domain. Due to the specific channel access technology depicted in Table 2, the power consumption of V2X radios is usually high. Vulnerable road users usually carry user equipment with a limited battery capacity, which limits the usability of V2X radio chips. There are multiple approaches to overcome this issue. One of them is to connect to RSUs via a different channel access technology, like Bluetooth, WiFi, or regular cellular communication, and ask the RSU to communicate the information from the VRU. Another, also promising, technique is to use the V2X radios in transmission-only mode. This means that the radios can save battery life but reception is not possible. The need for a GNSS signal in the case of C-V2X communication might also add further battery consumption issues.

#### 5.4.3. Tunnels and Urban Canyons—Use Cases without GNSS

Due to the specific channel access technology of C-V2X PC5, synchronization among the user equipment is required. In Mode 4/Mode 2 communication, it is solved by GNSS signals. Normally, depending on GNSS signal is not an issue since one of the most important pieces of information shared via V2X is the own position of the sender device, which comes from GNSS chips themselves. However, other positioning methods in the future might emerge, like ultra-wide band-based positioning, which could ensure the operation of V2X in such areas as well. However, V2X applications require relative position information. Relative position information could be valuable without a known absolute position as well. As we can see, dependency on GNSS signals might limit the usability of C-V2X technology in specific areas, like urban canyons or tunnels.

### 5.5. Positioning

In the safety-critical vehicular applications, as depicted in Section 2.4, the relative position of the traffic participants is the most interesting. The most obvious one could be the usage of various sensory data, like LiDARs or radars. The detected information can also be shared via standardized messages. There are several emerging technologies as well, like UWB [137], to obtain relative position data via radio communication. There are various efforts in 5G [21] to obtain positioning data. The following techniques support these efforts:Cell-identity-based;Angle-based;Range-based;Fingerprinting-based;5G-network-based positioning;Assisted positioning in 5G networks;Machine learning techniques.

The accuracy of these methods is promising. However, all of the mentioned techniques depend on 5G cells to calculate the position values, and we cannot depend on cell towers in the safety-relevant, direct communication use cases. If an ITS station is equipped with both 5G Uu and PC5, then these methods could be used for enhancing the quality of the ego position information, but, in order to obtain the relative positions among the stations, further developments should be made.

### 5.6. Architecture Obstacles and Challenges

#### 5.6.1. Cellular Communication (Uu Interface)—The Multiple MNO Problem

In the scope of 5G development, a lot of efforts were made to enhance the performance of cellular communication. Among other efforts, the latency of the communication was drastically decreased. The reduction in latency might open up the way to use the technology for V2X communication. One of the key advantages of this approach is that the infrastructure side of the communication—practically, the RSUs—is already deployed. Thanks to the novel multi-access mobile edge computing, virtualized RSUs could be used instead of expensive physical deployments. However, in the case of device-to-device communication use cases, the routing among the devices has to be solved. In this case, if the devices are subscribed to different mobile network operators, some of the advantages, like low latency, cannot be guaranteed anymore.

#### 5.6.2. Cellular Communication (Uu Interface)—Security Architecture and Compliance

In the V2X domain, trust among the vehicles is essential. This is ensured by using public key cryptography. The keys used to sign messages have to be well protected due to the safety aspects of V2X communication. Thus, private keys must be stored in hardware security modules in order to ensure that they are not leaked out. The virtualized environment used in the 5G systems, especially the MEC servers, provides numerous benefits for the applications, but the deployment of such hardware security modules and compliance with the strict security standards might be challenging. Another aspect of V2X communication is that, due to the distributed and localized broadcast properties of the communication, encryption is not facilitated. This approach is logical in the case of such a communication method, but it might not be sufficient in the case of centralized communication [14,17,25].

The evolution of cellular vehicle-to-everything (V2X) communication technology has encountered several obstacles and challenges that need to be addressed for its successful implementation. Ref. [58] emphasizes the need for continued research and development, as well as effective regulation, to ensure the successful implementation of these technologies.

#### 5.6.3. Lack of Backward Compatibility

As we could see, backward compatibility is not possible among Releases 14/15 and Release 16 for C-V2X. In cellular networks, this is not a huge issue since the base stations support both technologies, and the service is eventually handled on IP networks so that legacy devices can use the same services as the novel ones. Contrary to this, in the case of direct communication (for example, the CA service), an ITS station is a service provider and a client simultaneously. This means that services cannot move across the radio link technologies without a degradation in the service. In the case, for example, where we use the CA service with legacy ITS stations, the new ITS stations must use the same radio technology as the legacy nodes (or they need to use both the new and old radios), otherwise they cannot facilitate the other’s service.

#### 5.6.4. Mode 3 Communication

The C-V2X standards offer a specific, non-GNSS-signal-based way of operation, which promises to enhance channel utilization in a congested channel environment. However, this communication mode introduces a single point of failure, which is not acceptable in the case of safety-critical applications. A good question is who the operator of the orchestrator is, since it needs to be clarified who is allowed to provide such a service (just as its business model) [59].

## 6. 6G Visions for Overcoming the Obstacles

### 6.1. 6G Vision, Requirements, and Research

As the 6G roadmap for ITU and 5G-IA is in the “Vision” and “Requirements” phase [118] in 2023, and the specification phase of Release 20 will start in 2025 by 3GPP, there is room for optimist expectations.

The Next G Alliance has also provided a broad vision, roadmap, and development model on 6G, presenting the transatlantic perspectives in [119]. The six audacious goals of their vision are (i) sustainability; (ii) trust, security, and resilience; (iii) digital world experiences; (iv) cost-effective solutions; (v) distributed cloud and communications systems; and (vi) AI-native wireless networks. They list specific research challenges for each goal in [119].

To reach these goals, 5GIA grouped the envisaged key technologies [118] into the following categories: (i) system network architecture and control; (ii) edge and ubiquitous computing; (iii) radio technology and signal processing enhancements; (iv) optical network enhancements; (v) network and service security; (vi) non-terrestrial networks communication; (vii) special purpose networks/subnetworks; and (viii) opportunities for devices and components.

A conceptual solutions overview on integrated sensing and communications through dual-functional wireless networks for 6G is presented in [114].

The progress of the digital twin research and engineering domain affects V2X as well. The overarching study by Khan et. al on digital twins and 6G discusses various aspects from the architecture to DT deployment issues to the mobility management of edge-based twins [112].

### 6.2. Main Concepts Forseen for 6G-V2X

This section briefly summarizes the main concepts that are foreseen to affect 6G-V2x.

According to [8], there are technologies that are evolving in 5G and beyond and becoming mature for 6G, while others are revolutionary in the sense that their appearance will enable groundbreaking new results. The evolving technologies are the following:Hybrid RF-VLC V2X—providing higher data rates and demanding a low setup cost;Multiple radio access technologies—having the benefits of sub-6 GHz carriers but with a longer range;Non-orthogonal multiple access (NOMA)—allowing the combination of massive connectivity with ultra-low latency;The new multicarrier scheme—for higher spectrum and power effectiveness, even against the Doppler effect;Advanced resource allocation—for enhanced context awareness and to support cross-layer resource allocation;Integrated sensing, localization, and communication—for improved situation awareness;Satellite/UAV-aided V2X—as one of the common expectations for 6G infrastructure, to provide flexible, aerial base stations and wider coverage;Integrated computing—faster computing with lower operational costs;Integrated control and communications—the co-design of control and communications.

There are revolutionary technologies that are quite mature [8], so they could become an important part of 6G-based ecosystems:Blockchain—as it could significantly enhance security for some services;ML-aided V2X design—as it could provide significant performance enhancement for highly adaptive and complex V2X environments.

Furthermore, there are some revolutionary technologies—called revolutionary for 6G by [8]—that are less mature but might still be considered to support 6G:A programmable V2X environment—especially for enhancing the robustness and resilience of the radio interface;Tactile communication—as real-time haptic information transmission is one of the generic expectations for 6G;Quantum computing—as the ultimate superior computation mechanism that also enhances security;Brain–vehicle interfacing—as brain communication interfacing evolves, it could enable various supporting functions;THz communications—allowing an extremely high throughput.

Some research areas are receiving specific attention. Offloading and resource management is an important aspect; hence, [117] proposed a federated learning empowered computation offloading and resource management (FLOR) method, which offloads and manages the resources with low latency. Another aspect is data privacy, as information is passed among V2X users, often in open channels of heterogeneous technologies. Certain security and privacy issues can be handled by blockchain technologies, and processes can be optimized by federated learning (FL) at the edge. Hence, the BFLEdge [113] framework is proposed to cover the gap between these technologies. Positioning for V2X is also a key issue; advances from 5G to 6G are discussed in [21].

Although our current article does not focus on UAVs, 6G is expected to solve various issues around UAV control, communication, and management. Survey papers are already available for the anticipated 5G to 6G visions for UAV communication [115] and for green UAV communication for 6G [116].

## 7. Conclusions

V2X is a promising new technology, which aims to solve many transportation-related problems, but, first of all, it tries to create accident-free transportation. As we can see, this type of communication has a lot of peculiarities, which the radio access technology has to reflect on. We mainly focused on the direct V2X communication due to its high acceptance among OEMs and the fact that these are typically standardized messages with well-defined service operations and message formats. In the presented research work, we wanted to highlight the importance of domain-specific optimization and use-case-based evaluation, considering the special requirements of direct V2X communication. The most important requirements, where we found controversies and obstacles, can be found below:The communication range.Security flaws, including passive attacks, pilot contamination, MCS saturation, uplink jamming, leaky wave antennas, and the physical layer key.Used frequencies.Positioning.Decentralized network (having multiple MNOs can cause issues).A dependency on third-party services (the lack of GNSS signal can result in service degradation).Backward compatibility.

As part of future research, the found controversies could be further evaluated numerically based on the use cases. These controversies can also lay the foundation for further research to eliminate the technology’s existing weaknesses. The practical application of the identified problems can differ considerably. Some of the problems have initial solutions, such as local positioning or time-sensitive use cases; others do not. Moreover, there are problems, such as the multiple mobile network operator or the used frequencies problem, where the primary problem is not mainly technological but regulatory or financial.

Many of these issues are addressed by mid- and long-term solutions for 5G advanced recommendations and 6G visions. This paper provides a brief overview of these activities and some proposals that are highly relevant for C-V2X.

## Figures and Tables

**Figure 1 sensors-23-07031-f001:**
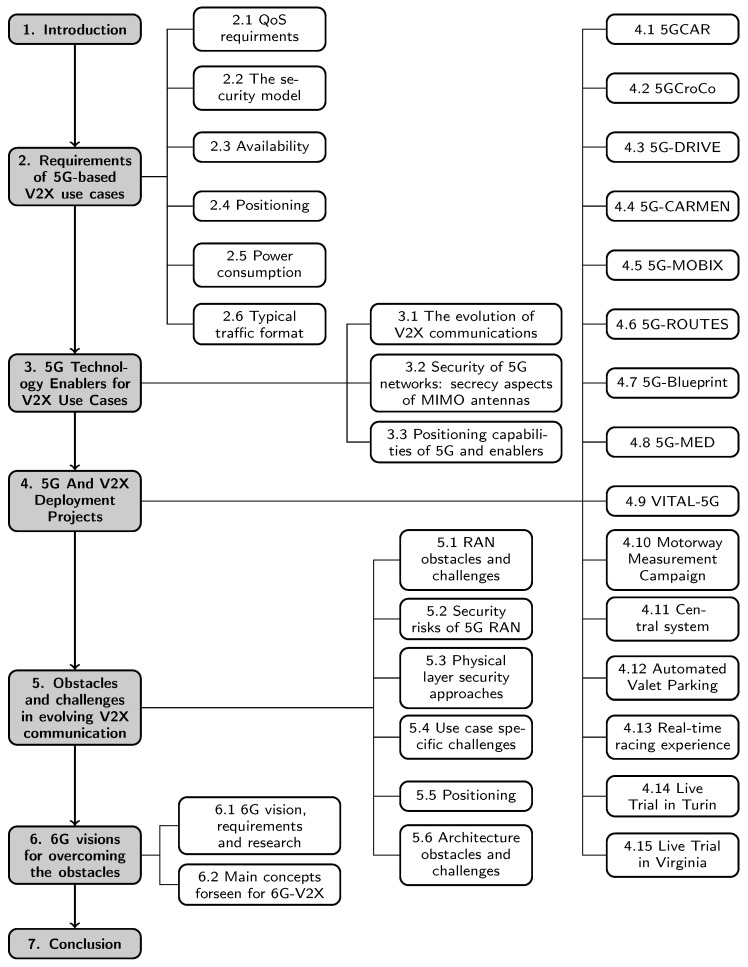
The structure of this paper.

**Figure 2 sensors-23-07031-f002:**
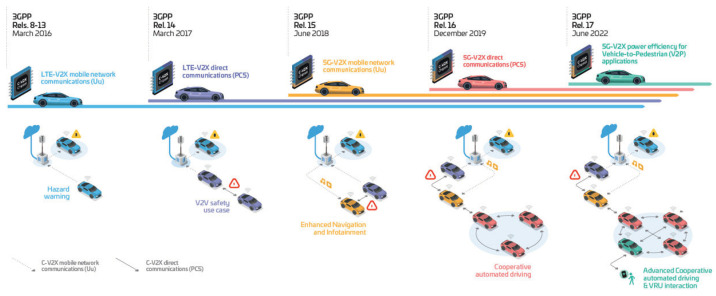
Timeline of the 3GPP releases, their main features, and the related V2X use cases [121].

**Figure 3 sensors-23-07031-f003:**
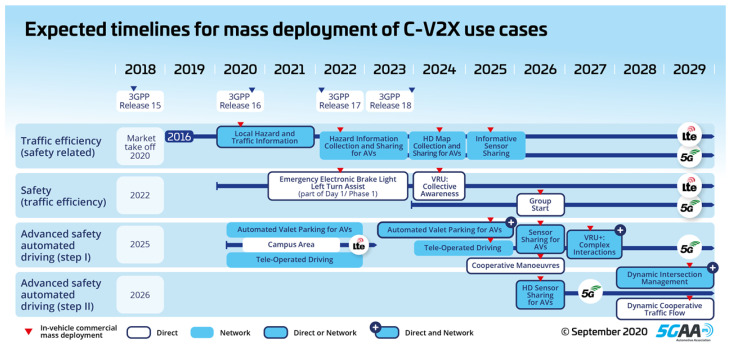
C-V2X deployment timeline for different types of use case groups [121].

**Figure 4 sensors-23-07031-f004:**
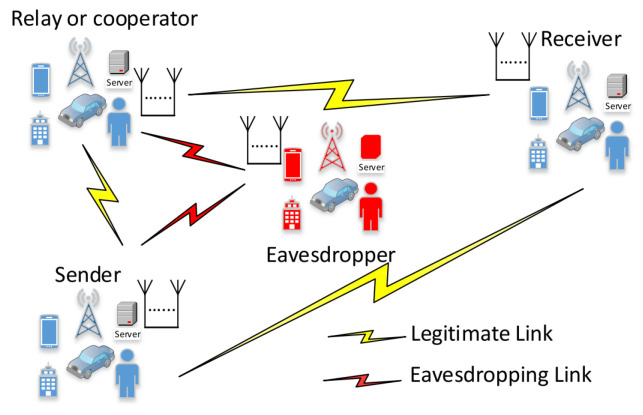
Eavesdropping model [126].

**Figure 5 sensors-23-07031-f005:**
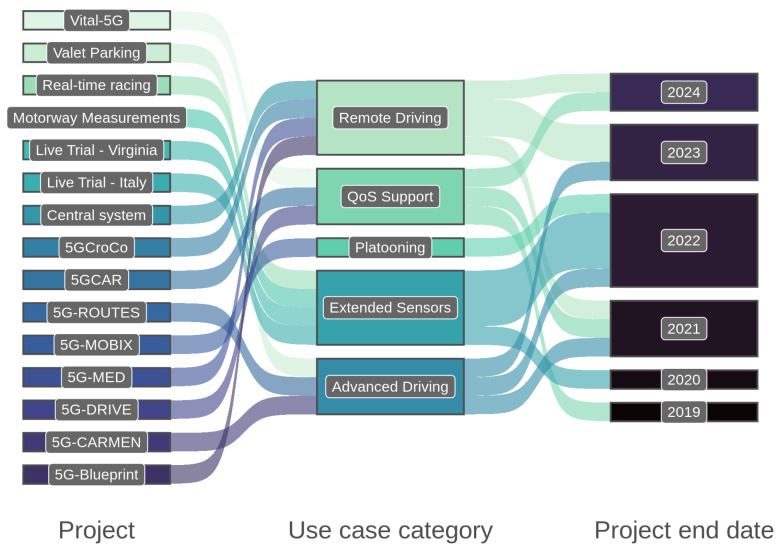
Projects’ end dates and use case focus based on the 3GPP V2X use case scenarios.

**Figure 6 sensors-23-07031-f006:**
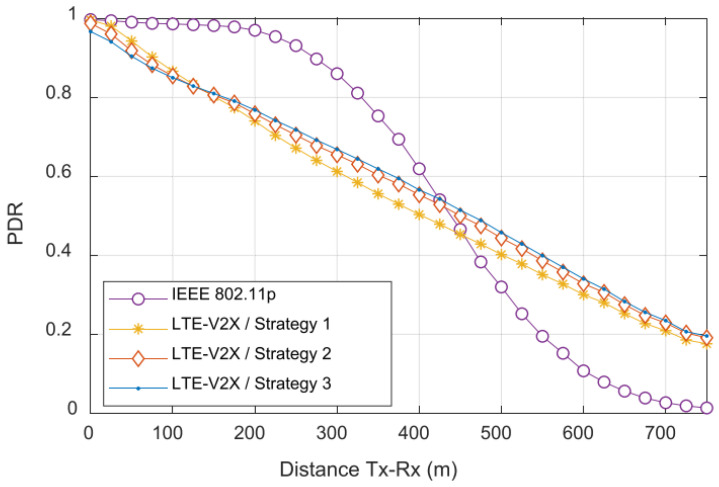
Packet Delivery Ratio experienced with aperiodic messages of variable size (empirical CAM model) [135].

**Figure 7 sensors-23-07031-f007:**
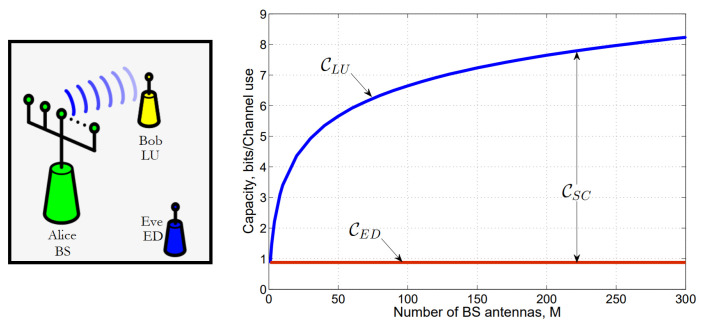
Change in secrecy capacity by the number of base station (BS) antennas in case of passive eavesdropping [111]. The secrecy capacity of Eve remains constant as the MIMO system focuses a beam on Bob.

**Table 1 sensors-23-07031-t001:** The summary of the used references grouped by document type and topic.

Topic	Document Type	References
Related works	Scientific publication	[6,7,8,9,10,11,12,13,14,15,16,17,18,19,20,21,22,23,24,25,26]
	Project report	[1,2,3,4]
	Standard	[5,27,28,29,30]
Technology enablers	Scientific publication	[8,16,31,32,33,34,35,36,37,38,39,40,41,42,43,44,45,46,47,48,49,50,51,52,53,54,55,56,57,58,59,60,61,62,63,64,65,66,67,68,69,70,71,72,73]
	Standard	[5,28,30,74,75,76,77,78,79,80,81,82]
V2X deployment projects	Scientific publication	[7,83,84]
	Study, project report	[85,86,87,88,89,90,91,92,93,94,95,96,97,98,99]
Obstacles	Scientific publication	[21,58,100,101,102,103,104,105,106,107,108,109,110,111]
6G vision	Scientific publication	[8,21,112,113,114,115,116,117]
	Study, project report	[118,119]

**Table 2 sensors-23-07031-t002:** Comparison of RAN properties of the direct communication V2X technologies.

Property	Release 14/15 PC5	Release 16 PC5	802.11p
Modulation	SC-FDM	OFDM	OFDM
Time synchronization	Tight synchronous	Tight synchronous	Loose asynchronous
Transmission scheduling	Semi-persistent sensing of least-occupied resource	Semi-persistent sensing of least-occupied resource	CSMA
Transmission time	1 ms	1 ms	Depends on packet length (0.4 ms typically)
Line coding	Turbo	Turbo	Convolutional
Operation frequency	5.9 GHz	5.9 GHz	5.9 GHz

**Table 3 sensors-23-07031-t003:** Comparison table for positioning technologies over 5G. The respective abbreviations are the following: least squares (LS), line of sight (LOS), non-line of sight (NLOS), extended Kalman filter (EKF), round trip time (RTT), received signal strength (RSS), angle of departure (AOD), angle of arrival (AOA), direction of arrival (DOA), time of arrival (TOA), time difference of arrival (TDOA), observed time difference of arrival (OTDOA), multiple signal classification (MUSIC), non-line of sight multiple signal cancellation (NC-MUSIC).

Name	Bases of Method	Advantages	Disadvantages	Further Improvements	Ref.
RTT	time-based	errors by sync can be ignored	sensitive to ranging errors	multi-cell RTT	[31,43]
RSS	range-based	low complexity	inaccurate (meter-level)	-	[38,50]
AOD	angle-based	simple calc. method (LS)	error source: NLOS	downlink AOD	[43,50]
AOA (DOA)	angle-based	simple calc. method (LS)	LOS condition needed	uplink AOA, large antenna arrays	[31,36,41,43,47,50]
TOA	range-based, spherical	simple	clock drift, noise, fading, and Doppler shifts	energy detector [44,45], correlation receiver [44,45]	[16,36,38,50]
TDOA	time difference (range-based)	no need for synchronization	noise-sensitive	applying equivalent FIM	[41,43,44,50]
OTDOA	reference signal time difference	commonly deployed	accurate time-delay estimation needed	design options are open	[31,43]
MUSIC	multiple signal classification based on AOA measurements	EKF-based, high accuracy	complex	NC-MUSIC	[31,43]

**Table 4 sensors-23-07031-t004:** Comparison table for hybrid localization technologies. The respective abbreviations are the following: inertial measurement unit (IMU), simultaneous localization and mapping (SLAM), Poisson multi-Bernoulli mixture (PMBM), device-to-device (D2D), cumulative density function (CDF), asymmetric visibility (AC), urban canyons (UC), Cramer–Rao bound (CRB), Fisher information matrix (FIM), weighted least squares (WLS).

Technologies	Main Approach	Methods	Results	Comment	Ref.
IMU—5G	edge computing	EKF for fusing the estimation results	a simulation environment for scene generation, signal propagation, and position estimation	submeter accuracy, the effect of the number of base stations tested	[38]
SLAM—5G	intermediate approach (the wave is not directly mapped to the position)	end-to-end processing chain, PMBM filter	problem decomposition with real channel estimation	mapping and vehicle state estimation handled simultaneously, accurate estimated number, type, and position of landmarks	[48]
GNSS—5G	newly proposed method: crossover multiple-way ranging (CO-MWR)	D2D range and angle measurements, integrated algorithms, state dimension reduction	fewer communication resources needed	less computation without losing information, accurate and robust method	[49]
GNSS—5G	physical-layer abstraction-based simulation, WLS algorithm	urban macro-cell environments, Gaussian-distributed error model, CDF	different GNSS constellations examined	5 m horizontal accuracy in 95% of cases in not ideal conditions (AV, UC)	[44]
GNSS—5G	visibility mask handling cmWave and mmWave signals	evaluation: CRB, FIM, number of base stations, CDF	simulation framework presented to test typical urban scenarios	NLOS signals have to be weighted, submeter accuracy achieved	[39]

**Table 5 sensors-23-07031-t005:** Comparison table for cellular V2X projects from the recent years.

Project Name	Project Life	Sites	Focus
5GCAR [129,130,131]	2017–2019	France	- 5G V2X architecture with IoT platform
5GCroCo [85,88]	2018–2021	France, Germany, and Luxembourg	- CCAM in the cross-border corridor - Predictive QoS
5G-DRIVE [132]	2018–2021	Finland, Italy, and UK (new)	- EU–China collaboration development based on 5G
5G-CARMEN [89]	2018–2021	Germany, Austria, and Itay	- Enabling self-driving vehicles - Multi-tenant platform
5G-MOBIX [90]	2018–2022	Spain–Portugal	- Evaluating connected and automated mobility applications
Motorway Measurement Campaign [84]	2020	Hungary	- Road geometry mapping and digital twin
Central system for supporting automated vehicle testing and operation [91]	2021–2024	Hungary–Austria	- Autonomous vehicles in cooperation with infrastructure
5G-ROUTES [92]	2020–2023	Latvia–Estonia–Finland	- Deployment of 5G end-to-end interoperable CAM services
5G-Blueprint [93]	2020–2023	Belgium–Netherlands	- Evaluating cross-border teleoperated transport
5G-MED [94]	2020–2023	Spain–France	- 5G infrastructure architecture for roads and railways
Vital-5G [95]	2021–2024	Romania–Belgium–Greece	- Network applications for the T and L industry
Automated valet parking over 5G Network [96]	2022	Germany	- Automated valet parking
Real-time racing experience on 5G network [97]	2022	Hungary	- Real-time digital twin on race track
Live trial of 5G-connected car concept to launch in Turin, Italy [98]	2022	Italy	- Connected services to the digital transformation of smart cities
Live trial of 5G-connected car concept launches in Blacksburg, Virginia [99]	2022	USA	- Connected mobility to enhance safety on roads.

**Table 6 sensors-23-07031-t006:** Summary of the flaws, available solutions, and severity of the problem (Severity/Importance (S/I): Critical (C), High (H), Medium (M), Low (L)).

Method	Severity/Importance	Solutions/Advantages
Passive attacks	Medium	Increase the No. of antennas, beam focusing
Pilot contamination	High	Encrypted or/and scalable pilot set
MCS saturation	Medium	Appropriate power control
Uplink jamming	Critical	Allocating antennas to jam Eve, increase the No. of antennas
Leaky wave antennas	High	Unique properties, increased secrecy capacity
Physical layer key	High	Low need for computation resources

## Data Availability

Not applicable.

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
