# Peer review of "Large-Scale Cellular Vehicle-to-Everything Deployments Based on 5G—Critical Challenges, Solutions, and Vision towards 6G: A Survey"

_sensors, 2023, doi:10.3390/s23167031_

Round 1

Reviewer 1 Report

First of all, I would like to acknowledge that the topic of your manuscript, "Large-Scale C-V2X Deployments Based on 5G - Critical Challenges, Solutions, and vision towards 6G", is very timely and undoubtedly of great interest to many readers in this field. The relevance of the subject matter is undoubtedly a strength of this paper.

However, I believe a few areas could be improved to strengthen the manuscript further. To begin with, the treatment of the requirements, technology enablers, and projects related to C-V2X seem to be largely derived from standardization documents and white papers. While it is important to provide readers with a thorough background, it may also be beneficial to delve beyond the surface and discuss these topics more in-depth. This could involve providing a unique analysis or utilizing a novel methodology to shed new light on these topics.

Related to this, all figures in your manuscript appear to be cited from other papers. While using existing figures for illustrative purposes is not inherently problematic, it might be beneficial to include more original figures or diagrams to provide a fresh perspective or a deeper understanding of the topic.

Another point of consideration is the framing of the paper itself. As it currently stands, it seems more like a survey paper, despite the title not explicitly indicating so. Readers might misunderstand the article's nature, expecting it to be a traditional academic paper presenting new technologies enabling large-scale C-V2X deployments. To alleviate potential confusion, it might be beneficial to make it clearer in the title or the abstract that this is a survey or review paper, summarizing the current state of the field rather than introducing new methodologies or analyses.

The quality of English Language is good. 

Reviewer 2 Report

The topic is interesting and current. The paper is well-conceived and written. However, before publishing, I suggest a minor revision.

Several recent references are missing in the introductory part.

I ask you to describe the objectives listed in point 4.11 in more detail.

The methodology is clearly presented.

I ask you to expand the conclusion section with the limitations of the study, future directions of research, and application of the obtained results in practice.

Reviewer 3 Report

The authors analyzed the key challenges associated with the large-scale deployment of 5G-based C-V2X systems. In addition, they addressed obstacles and possible contradictions in the C-V2X standards caused by the special requirements. My comment to the authors as follows:

1) At the end of section 1 and before last paragraph, authors are required to rewrite the contributions in very clear miner without repeating the sentences. I suggest to give each contribution numbered list.  

2) In section 2; Requirements of 5G-based V2X use cases: the 2.1 and 2.2. can be discussed as Quality of services which includes four parameters Latency, Jitter Delay, Packet Drop, and Throughput. So, authors are advised to explain the QoS not only latency and packets drop.    

3) Authors are required to add a well discussion related works before section 2.    

4) Authors are required to explain security/cybersecurity requirements in more details. They should add man-in-the-middle attacks, Denial of Service, etc.

5) Some references are annual version such as ref[1],[2], and [3], can you use the lates one only.  

Three are many typos mistakes in the paper. Authors are required to enhance some sentences and remove the typo mistakes (e.g., in line 322 3GPP 6G release –, which)

Round 2

Reviewer 1 Report

The paper provides a comprehensive discussion of challenges and solutions related to the deployment of Cellular Vehicle-to-Everything (C-V2X) systems. It offers valuable perspectives for various stakeholders and successfully bridges theoretical understanding and applied practice. Although the paper is well-researched and well-structured, I suggest the authors in their future works could, delve deeper into potential deployment strategies for C-V2X and offer a more detailed comparison of how 5G and 6G may handle these systems differently. These are not critiques but suggestions for future research. However, in its current form, I highly recommend the paper for publication.

Good enough. 

Author Response

We thank the valuable suggestions and critical assessment of our work. We will act considering these aspects in the future.

Reviewer 3 Report

The paper (Manuscript ID: sensors-2437251is a review paper, so the summarized table of related works must be included in section 1 or 2.

-

Author Response

Thank you for the comment, we have included a table to summarize the used references.

Round 3

Reviewer 3 Report

Authors addressed all my comments.